# The Use of Wavelet Analysis to Improve the Accuracy of Pavement Layer Thickness Estimation Based on Amplitudes of Electromagnetic Waves

**DOI:** 10.3390/ma13143214

**Published:** 2020-07-19

**Authors:** Małgorzata Wutke, Anna Lejzerowicz, Andrzej Garbacz

**Affiliations:** 1TPA Sp. z o.o., Parzniewska 8, 05-800 Pruszków, Poland; 2Faculty of Civil Engineering, Warsaw University of Technology, Al. Armii Ludowej 16, 00-637 Warsaw, Poland; a.lejzerowicz@il.pw.edu.pl (A.L.); a.garbacz@il.pw.edu.pl (A.G.)

**Keywords:** ground penetrating radar (GPR), HMA dielectric constant, wavelet analysis, road pavement thickness estimation

## Abstract

The article discusses one of the methods of dielectric constant determination in a continuous way, which is the determination of its value based on the amplitude of the wave reflected from the surface. Based on tests performed on model asphalt slabs, it was presented how the value of the dielectric constant changes depending on the atmospheric conditions of the measured surface (dry, covered with water film, covered with ice, covered with snow, covered with de-icing salt). Coefficients correcting dielectric constants of hot mix asphalt (HMA) determined in various surface atmospheric conditions were introduced. It was proposed to determine the atmospheric conditions of the pavement with the use of wavelet analysis in order to choose the proper dielectric constant correction coefficient and therefore improve the accuracy of the pavement layer thickness estimation based on the ground penetrating radar (GPR) method.

## 1. Introduction

The requirements for pavements, not only newly built, but also existing, maintained, and undergoing renovation are increasing along with new guidelines [1]. The proper recognition of the thickness of pavements layers [2,3] directly influences the quality assurance of assessing the current pavement load capacity and the correctness of the pavement repair technology being developed.

The measurement of road pavement thickness can be performed using non-destructive methods such as the ground penetrating radar (GPR) method [4,5,6,7,8,9], eddy currents (StratoTest device) [10], Impact Echo [11,12,13,14], and ultrasonic [15] or semi-destructive [16] method, which is drilling. However, the eddy current method is only suitable for measuring the thickness of new surfaces, since it requires placing an aluminum reflector on the bottom of the layer [10]. The Impact Echo method is used only to measure thick elements, and repaired surfaces often have a thickness smaller than 10 cm [12]. While the Impact Echo device has been developed that has many probes that can be implemented to study large areas [14], the device based on ultrasound allows only point measurements [15]. In addition, the high frequency of ultrasonic devices is associated with attenuation of the signal in shallow layers, which means that the signal may not reach the bottom of the layer and thickness measurement may not be possible. Therefore, it is concluded that GPR is the best device for the continuous measurement of road surface thickness.

Identifying the thickness of 10 km road pavement layers based on boreholes is almost twice as expensive as that based on GPR surveys. In addition to the price, the social burden and destructive interference in the pavement is increasing the likelihood of premature degradation; however, when the structure is recognized based on GPR measurement, these negative consequences are much smaller. In addition, the detail of road construction information obtained from GPR measurement is incomparable to the information obtained from boreholes. A better diagnostic of road structures means, first, a greater safety of road users, and additionally, a possible reduction of costs of renovation works. Continuous recognition of the pavement structure enables an optimal and proper design of renovation treatments. By reducing the thickness of the layer by 1 cm on a 100 km stretch of road consisting of two lanes and a roadside, according to the present prices, we can save around PLN 600,000–800,000 on the asphalt wearing course and PLN 400,000–500,000 on the asphalt bonding and the asphalt base. This prompts an in-depth analysis of the issue of determining the thickness of road pavement layers using the GPR method.

The aim of this paper is an estimation of pavement thickness with GPR based on advanced signal analysis using wavelet transform.

## 2. Pavement Layers Thickness Estimation Based on GPR Method

### 2.1. Thickness Estimation Based on GPR Method

To calculate the layer thickness based on the GPR test, the following equation is used [5]:(1)$$d=\frac{c\cdot t}{2\cdot \sqrt{\varepsilon_{r}}}$$
where:
d [cm]layer thicknessc [cm/ns]speed of light propagation in a vacuum, c=30 cm/ns
t [ns]two-way travel time between reflections from boundary surfaces$\varepsilon_{r}\;\left[-\right]$dielectric constant of the layer.

The unknown in Equation (1) is the dielectric constant of the medium, which depends on many factors, including among others the humidity of the medium, porosity, and mineral composition. The dielectric constant can be selected based on the values published in the literature (rough information, which is shown in Table 1 and Figure 1), which may be calculated based on a drilled core (accurate, but point information) or calculated based on the amplitude of the wave reflected from the surface (the value depends on the surface conditions).

### 2.2. Dielectric Constant of Hot Mix Asphalt (HMA) Published in Literature

The wide range of values of the dielectric constant of the pavement layers, which are taken for the thickness determinations, causes errors. Table 1 summarizes the dielectric constants of bitumen determined in microwave frequencies given in various sources. For example, according to [17], the range of dielectric constants of asphalt mixtures determined by the GPR is from 2 to 12, according to [18] from 2 to 4, according to [19] from 4 to 10, and according to [20] from 4 to 15.

As one can see then, the dielectric constants of materials published in the literature should be used only as an indication of the approximate value of the electrical properties of the tested medium. A way to accurately determine the dielectric constant is to drill a core and calculate it based on the thickness of the core. However, as a result, the dielectric constant value is obtained at one specific point, and its value can also change between drilling points, especially in the case of roads after repeated repairs. In addition, we strive to interfere as little as possible in a destructive way, which is drilling, into the conditions on the pavement and reduce its durability. To correctly determine the thickness of the medium by the GPR method, the dielectric constant should be determined continuously and in a completely non-destructive way.

### 2.3. Calculation of Dielectric Constants Based on the Amplitude of the Wave Reflected from the Surface

The method of determining the dielectric constants of a medium in a continuous way is to calculate it based on the amplitude of the wave reflected from the surface and the amplitude of the wave reflected from the ideal reflector, e.g., metal plate (reference amplitude). In this case, the following formula is used [5,6]:(2)$$\varepsilon_{r}={\left[\frac{1+\frac{A_{0}}{A_{m}}}{1-\frac{A_{0}}{A_{m}}}\right]}^{2}$$
where:
$\varepsilon_{r}\;\left[-\right]$dielectric constant of the first layer of the medium$A_{0}\;\left[-\right]$reflected wave amplitude on the border: air-tested surface$A_{m}\;\left[-\right]$reflected wave amplitude on the border: air-metal plate (reference amplitude).

However, as already mentioned, the result of determining the dielectric constant, as it is calculated based on the amplitude of the wave reflected from the surface, strictly depends on the conditions on the surface. In the presence of a water film on the surface, dielectric constants calculated based on the amplitude of the wave reflected from the surface increase, which was confirmed by research by the authors of this publication in the paper [24] (as a result of the presence of a water film on the asphalt pavement resulting from a small rainfall, an increase in dielectric constant from about 6, determined on a dry surface, to 9 determined on a wet surface was observed).

### 2.4. Impact of Incorrect Estimation of the Dielectric Constant on the Accuracy of Asphalt Pavement Thickness Determination

Figure 1 shows how an incorrect determination of the dielectric constant affects the error in measuring the asphalt pavement thickness using the GPR method. A surface made of HMA with a dielectric constant of 5 and thickness of 22 cm is assumed. Then, the two-way travel time is 3.3 ns (nanoseconds). Assuming such a time and taking the dielectric constants given in the literature from 2 to 15 [18,20] for calculations, the relative error in thickness estimation is determined as a commonly used formula [25]:(3)$$\Delta d=\frac{d-d_{ref}}{d_{ref}}\cdot 100\%$$
where:
Δd [%]relative error in thickness determination using the GPR methodd [cm]layer thickness determined by the GPR method$d_{ref}\;\left[cm\right]$reference thickness of the layer (thickness of the drilled core).

The relative error in thickness determination using the GPR method is from −58% (for a dielectric constant of 2) to +42% (for a dielectric constant of 15). It is noted that the assumption of a smaller dielectric constant than the real one has a stronger impact on the actual thickness value than using higher dielectric constant values.

The article proposes the use of one of the methods of advanced GPR signal analysis—wavelet analysis—as a support tool in determining the atmospheric conditions on the surface of the HMA during GPR tests (in determining whether the HMA surface was dry, wet, covered with ice, snow or de-icing salt). Depending on the surface conditions, the dielectric constants will be corrected accordingly.

## 3. Wavelet Analysis of GPR Signal

### 3.1. Theoretical Foundations of Wavelet Analysis

Wavelet analysis is based on wavelet functions—the equivalents of trigonometric functions found in the Fourier analysis [26]. Wavelet analysis splits the signal into components that are properly scaled and shifted (in time) by the basic wavelet (mother wavelet). Unlike trigonometric functions, wavelet functions are not periodic—they are irregular and asymmetrical, and their shape is similar to the shape of the pulse emitted by the radar antenna.

The idea of the wavelet transform is similar to that of the Fourier transform—they both rely on the decomposition of the examined signal into component functions, but instead of the harmonic components of the Fourier transform, wavelet functions of different scale and position are used. Scaling a wavelet means stretching or compressing it. Moving the wavelet, as the name implies, is a change in the position of the wavelet on the time axis. The results of the wavelet analysis are the wavelet coefficients, which are the sum relative to the time of the product of the signal and the scaled and shifted forms of the wavelet.

Wavelet coefficients describe how the wavelet function of a certain scale and position is similar to the signal fragment being considered. A smaller scale factor corresponds to a more ‘compressed’ wavelet. Larger scale values correspond to more ‘stretched’ wavelets. The more stretched the wavelet, the longer the signal range with which it is compared, and therefore the rougher the signal features represented by the wavelet coefficients that are obtained. The results are presented in the form of a wavelet scalogram showing the energy distribution of the signal in the coordinates time (shift)-scale [27]. It is the distribution of energy into individual wavelet coefficients as a function of scale and time. The scale allows determining what spectra the GPR pulse has at different time intervals (at different depths).

The results of the wavelet analysis depend on the type of the wavelet we choose as the wavelet mother. MatLab is a popular tool for wavelet analysis. The following wavelets can be used: Daubechies, Coiflet, Gaussian derivative, Haar, Symlets, Biortogonal, reverse Biortogonal, Meyer, discrete approximation of Meyer, Mexican hat, Morlet. The wavelet sets that can be used are Gaussian derivative, Shannon, and comprehensive Morlet. The wavelets that are the most often used in GPR signal analysis—Daubechies 4, 5 and 6, Bioorthogonal 3, 3.1, 3.5, and 7, and Symlet 6—will be discussed in more detail in the next section.

### 3.2. The Use of Wavelet Analysis in GPR Signal Interpretation

Reviewing the applications of wavelet analysis as a tool for processing signals emitted by non-destructive diagnostics devices mostly leads to monitoring the condition of engineering structures or its elements. Publication [28] presents the possibility of using wavelet analysis to monitor the condition of composite panels. It has been observed on wavelet spectrograms from various plates that the time–frequency structures differ in the dominant frequency bands depending on the moisture content of each panel. Publication [29] presents an application of continuous wavelet transform in vibration-based damage detection method for beams and plates. The promising application of a wavelet analysis of Impact Echo signals to assess the quality of concrete structure repairs is presented in publication [30].

The scope of applications of wavelet analysis of GPR signals in road infrastructure includes the use of wavelet analysis, among others, in the assessment of the backfill of the tunnel [31]. The tests were carried out using a 400 MHz ground-coupled antenna, and the GPR signal was compared to the Daubechies 4 wavelet. After analyzing the A-scan from a place where clays, silt, and water occur, faster suppression of the high-frequency component and slower suppression of the low-frequency component were observed.

Another application of wavelet analysis is the assessment of the condition of railway ballast in terms of the presence of impurities in it [32]. The standard deviations of the wavelet coefficients were considered to represent the signal scattering intensity. It turned out that the standard deviation decreases as the level of pollution increases. The measurements were made with an air-coupled 2.0 GHz antenna; 5 levels of signal decomposition with a Daubechies mother wave were used.

Wavelet analysis is also used to remove noise and interference, especially where the useful signal-to-noise ratio is so small that the correct signal is not visible. In measurements taken with 25, 50, 250, and 500 MHz ground-coupled antennas to distinguish geological layers, wavelets Bioorthogonal 3.1 (for 25 and 50 MHz antennas), Bioorthogonal 3.5, 3 (250 MHz), and 7 (500 MHz) were used. Noise reduction was based on 5-level decomposition of the recorded GPR signal [33]. In [34], it was checked which wavelets are best suited for removing noise from the GPR signal from measurement on a flexible surface using a 1 GHz antenna. Daubechies 6, Symlet 6, Biorthogonal, and Haar wavelets were tested (the Haar wavelet is significantly different from the GPR signal; this served to indicate the essence of the appropriate selection of the mother wavelet and to highlight the effect of this choice on the result); Daubechies 6 and Symlet 6 wavelets turned out to be the best for this application.

## 4. GPR Measurements of Asphalt Slabs in Various Atmospheric Conditions

### 4.1. Testing Area

Table 2 shows different constructions of tested model asphalt slabs with dimensions of 50 cm × 50 cm × 22 cm. Each slab consists of three layers with a total thickness of 22 cm. The slabs differ in their wearing course type, which is: MA—mastic asphalt being the most tight asphalt mix, SMA—stone mastic asphalt, AC—asphalt concrete, BBTM (*fr. beton bitumineuse trés mince*)—asphalt concrete for thin layers, and PA—porous asphalt, being the most porous asphalt mix. The slabs were placed on a metal plate to intensify wave reflection from the bottom of the slab (see Figure 2). The metal plate has a dielectric constant higher than asphalt; hence, during propagation by asphalt media, the wave phase does not change. This is important from the point of view of marking the bottom level of the slab.

Table 3 shows the atmospheric conditions when performing different GPR tests. The reference measurement was made at 28 °C (wI). Other measurements were made at a temperature below zero in the following order: on the dry surface of the slab (wII), after pouring water onto the slab (wIII), after ice formation on the slab surface (wIV), in the presence of a thin layer of fluffy snow on the slab (wV), and in the presence of de-icing salt on the surface of the slab (wVI). Measurements were carried out using a GSSI (Geophysical Survey Systems, Inc., Nashua, NH, USA) air-coupled antenna with a central frequency of 1.0 GHz.

### 4.2. A-Scans from m1–m5 Slabs Measurements in Various Atmospheric Conditions

In Figure 3, Figure 4 and Figure 5, A-scans from measurements of slabs m1–m5 in conditions wI–wVI are shown. As a zero level, the minimum amplitude of the wave reflected from the slab surface was assumed. The bottom level is the minimum amplitude of the wave reflected from the bottom of the metal plate under the asphalt slab. Based on the A-scans, it is visible that both the propagation time through the slab and the reflection amplitude from the surface vary depending on the weather conditions.

### 4.3. Dielectric Constants of HMA

#### 4.3.1. Dielectric Constants Calculated based on Propagation Time through the Slab of Known Thickness, Hereinafter Named the “B” Method

Table 4 shows two-way travel time through the slabs m1–m5 read directly from the GPR measurement. Knowing that the thickness of slabs m1–m5 is equal to 22 cm, and transforming Equation (1), their dielectric constants ($\varepsilon_{r_{B}}$) were calculated. The results are summarized in Table 5.

The differences in dielectric constants depending on the atmospheric conditions during the measurement calculated by the “B” method are not large (Figure 6), but they cannot be ignored. The following trends of apparent increases and decreases of the dielectric constant of the HMA are noted: temperature below zero (wII) causes an apparent decrease of the dielectric constant determined by the “B” method. Pouring the slab with water (wIII) causes an apparent increase in the dielectric constant. Freezing of the formed water film (wIV) causes the apparent decrease of the dielectric constant determined by the “B” method. The presence of snow (wV) on the surface causes an apparent increase in dielectric constant (except slab m3). The presence of salt on the slab surface (wVI) also causes an apparent increase in HMA dielectric constant marked by method “B”.

For calculations of dielectric constants using the “B” method, knowledge of thickness is required, which in road practice translates into drilling and taking cores from roads for being measured. The following are dielectric constants calculated based on measurements of the same slabs, under the same atmospheric conditions, but based on the amplitude of the wave reflected from the surface, i.e., by a method that does not require taking cores.

#### 4.3.2. Dielectric Constants Calculated based on the Amplitude of the Wave Reflected from the Surface, Hereinafter Named the “A” Method

Calculations were made based on Equation (2). Table 6 shows the ratio of the wave amplitude reflected from the surface of the slab to the wave amplitude reflected from the metal plate (reference amplitude). Table 7 shows dielectric constants calculated based on amplitudes.

Based on the summaries in Table 7 and Figure 7, it is noted that the dielectric constants calculated based on the reflected wave amplitudes are significantly different from those calculated based on propagation time and known slab thickness. The temperature below zero (wII) of the HMA causes a significant decrease in the apparent value of its dielectric constant. Pouring the slab with water and the presence of a water film on the surface (wIII) causes an increase in dielectric constants calculated with the “A” method. Freezing of water on the surface (wIV) causes a decrease in dielectric constant again. The presence of snow (wV) decreases the dielectric constant (except for slab m5). The presence of salt on the surface (wVI) causes a significant increase in dielectric constant determined based on the amplitude of the wave reflected from the surface.

#### 4.3.3. Calculated Slabs Thicknesses Based on the Wave Amplitude Reflected from the Surface

Table 8 summarizes the thicknesses calculated based on dielectric constants determined based on the amplitudes of waves reflected from the surface. Table 9 shows the relative thickness determination error caused by calculating the dielectric constant based on amplitudes.

The measurement error in the conditions adopted as a reference is up to −10%, and it is the accuracy of determining the thickness of layers by the method based on the amplitudes of the wave reflected from the surface. Measurements in temperature below zero cause the calculated thickness to be smaller than the actual thickness up to −11%, except for slab m5. The presence of water film on the slab’s surface caused the calculated thickness to be up to −11% smaller than the actual thickness, except for slab m5. An icy slab surface caused the calculated thickness to be up to −7% smaller than the actual thickness, except for slab m5. The presence of snow on the slab surface caused the calculated thickness to be greater than the actual thickness of the slab (up to 10% greater), except for slab m5 (thickness −16% smaller). As a result of the decrease in freezing temperature during the presence of de-icing salt on the HMA surface of the asphalt mixture, apparently smaller thicknesses are obtained (up to −29%).

#### 4.3.4. Correction Coefficients for Dielectric Constants Determined based on the Amplitudes Reflected from the Surface

Based on Equation (4), the correction coefficients for dielectric constants determined based on amplitudes to dielectric constants determined based on propagation time through the slab of known thickness were calculated. Their values are summarized in Table 10. The reference conditions were the wI conditions (28 °C, dry slab surface). The dielectric constant determined based on amplitudes in all conditions other than wI should be multiplied by the correction coefficient k.
(4)$$k=\frac{\varepsilon_{r_{B}}}{\varepsilon_{r_{A}}}.\;$$
where:
k [−]correction coefficients for dielectric constants determined based on the amplitudes$\varepsilon_{r_{B}}\;\left[-\right]$dielectric constants calculated based on propagation time through the slabs (“B” method)$\varepsilon_{r_{A}}\;\left[-\right]$dielectric constants calculated based on the amplitudes (“A” method).

During determinations of the dielectric constant based on the amplitudes in a reference condition, a dielectric constant correction factor k = 0.87 should be adopted. During determinations of the dielectric constant based on the amplitudes at the temperature of −5 °C on the dry slab surface, a dielectric constant correction factor k = 0.97 should be adopted. During measurements at −5 °C and with water film presence on the slab surface, a dielectric constant correction coefficient k = 0.92 should be used. During measurements at −5 °C and with ice presence on the slab surface, a dielectric constant correction coefficient k = 0.98 should be used. During measurements at −2 °C and snow on the slab surface, a dielectric constant k = 1.02 correction coefficient should be adopted. During measurements at −2 °C and de-icing salt on the slab surface, a dielectric constant correction factor k = 0.55 should be used.

The thicknesses calculated based on dielectric constants corrected by the mentioned coefficients and the relative thickness measurement error are presented in Table 11 and Table 12, respectively.

It is noted that the thickness measurements error has been significantly reduced in the case of the slab covered with de-icing salt. Attention is drawn to the fact that the correction coefficients have been calculated as the average of the coefficients for 5 slabs with different HMA wearing courses for the determined atmospheric conditions. To significantly reduce the error, a factor calculated for a specific HMA wearing course should be applied.

The coefficients presented above are a proposal for specific slab geometry (m1–m5), specific measurement conditions (wI–wVI), and selected frequency and antenna design (1.0 GHz air-coupled). The observed trends thrive to expand the base of dielectric constants in controlled atmospheric conditions with a constant change in temperature, amount and type of precipitation, and the amount of de-icing salt used.

When measuring using GPR on large sections, the atmospheric conditions of the surface usually are not known; it is not known if and in which sections the surface was covered with water film, ice, snow, or de-icing salt. Of course, video cameras are helpful in recording the surface atmospheric condition; while it will be possible to record where the water film was on the surface during the measurements, the presence of salt on the surface will not be recognized based on the video image.

In determining the surface conditions, wavelet analysis may be a useful tool to correctly select the proposed correction coefficient.

### 4.4. Wavelet Analysis of the GPR Signal

Wavelet analysis was performed using the Db6 wavelet. The type of wavelet was chosen based on a literature review and the selection of the group of wavelets that are most useful in GPR signal analysis as well as the thorough initial empirical analysis of the authors.

Wavelet analysis was performed on the unscaled signal from m3 slab measurements. Figure 8, Figure 9 and Figure 10 compare the distribution of energy into individual wavelet coefficients as a function of scale and time. It was noticed that in some cases, the scale coefficients of signals obtained in different atmospheric conditions at a slab surface differ in magnitude. The largest magnitudes have signal scale factors from the measurement of a slab covered with de-icing salt, which is a positive result, because it is the presence of salt on the surface that strongly affects the value of the dielectric constant determined based on the amplitude, and, as previously noted, the presence of salt is not detectable by video recording. The results of the conducted analysis thrive at defining the atmospheric conditions on the surface (dry, covered with water film, ice, snow, and salt), which was obtained during the GPR survey.

There is a difference between the scalograms from the measurement signals taken at temperatures above zero (Figure 8a) and below zero (Figure 8b). Scale factors 9–60 at a distance of 180–250 samples have higher magnitudes when the measurement is performed at a temperature above zero. Scale factors 9–60 at a distance of 200–400 samples have higher magnitudes when the measurement is performed at a temperature below zero.

The scalograms from the measurement signals taken at temperatures below zero on a dry slab surface (Figure 8b), temperatures below zero and water film on the slab surface (Figure 9a), and temperatures below zero and a thin layer of ice on the slab surface (Figure 9b) do not differ significantly.

Snow on the pavement at a temperature below zero (Figure 10a) results in the fact that the majority of energy falls on scale factors 9–60 at a distance of 200–300 samples. The effect of salt (Figure 10b) on the surface is that the scale factors 9–60 at a distance of 0–150 samples have very large amplitudes: the largest of those obtained during all other surface conditions.

## 5. Discussion

Based on the research and analysis, it was found that the value of the dielectric constant determined continuously based on the amplitude of the wave reflected from the surface is affected by the atmospheric conditions on the surface—its temperature, moisture, ice or snow, and presence of de-icing salt.

When the temperature is high, HMA dielectric constants values are higher than at temperatures below zero. The observed tendency of the apparent increase in HMA dielectric constants with the increase in its temperature results from the more energetic movement of molecules—this enables an easier formation and orientation of dipoles and requires continuing testing on a larger number of samples and/or test sites in a continuous temperature range. As a result of the presence of a water film on the surface and partly in the pores of the HMA (water itself is a dipole), after the application of an external electric field, as expected, an increase in the polarization capacity of the medium was observed. The freezing of water on the surface and partly in the pores caused a slight decrease in the dielectric constant. This is due to the formation of hydrogen bonds and retention of the dipole polarization mechanism of water. An increase in dielectric constant, as it was observed, was due to the presence of snow on the surface. The apparent increase in HMA dielectric constant due to the presence of salt can be combined with the presence of Na^−^ and Cl^+^ ions as well as ionic polarization and increase in the material’s ability to compensate the electric field; on the other hand, the electric field between the Na^−^ and Cl^+^ ions and water molecules whose presence results from melting snow and ice on the surface of the slab should level out. This issue as well as the issue related to the presence of snow on the surface should be further investigated.

The measurement error in calculating the thickness based on reflection amplitude in the conditions adopted as a reference is up to −10%. The surface atmospheric conditions affect less or more the thickness value determined by the GPR method—and as a result of the thickness determination based on the dielectric constant calculated using the amplitude of the reflected wave from the surface below zero temperature, the thickness measurement error was up to −11 ÷ +8%; in the case of a wave amplitude reflected from the surface covered with water film, it was up to −11 ÷ +7%; in the case of a wave amplitude reflected from icy surfaces, it was up to −7 ÷ 8%; in the case of a wave amplitude reflected from the surfaces covered with a thin layer of powdery snow, it was up to −16 ÷ +10%, and in the case of a wave amplitude reflected from the surfaces covered with de-icing salt, it was up to −29% (see Table 9).

It was proposed to use advanced wavelet analysis to determine which atmospheric conditions on the surface of the pavement result in high/low dielectric constant values determined based on the amplitude. Promisingly, high differences in magnitudes of scale factors in relation to the reference conditions were obtained based on a signal from the measurements on the surface covered with de-icing salt, which is a very positive result, because the presence of salt on the surface is not observable on video records performed alongside GPR measurements.

For each of the analyzed atmospheric conditions, a correction factor for the dielectric constant calculated based on the wave amplitudes reflected from surfaces with different atmospheric conditions was proposed, whose value could be chosen based on wavelet analysis and exactly reflects the appearance of the wavelet scalogram (see Figure 8, Figure 9 and Figure 10). During determinations of the dielectric constant based on the amplitudes in a reference condition, a dielectric constant correction factor k = 0.87 should be adopted. During determinations of the dielectric constant based on the amplitudes at the temperature of −5 °C on the dry slab surface, a dielectric constant correction factor k = 0.97 should be adopted. During measurements at −5 °C and with a water film presence on the slab surface, a dielectric constant correction coefficient k = 0.92 should be used. During measurements at −5 °C and with ice presence on the slab surface, a dielectric constant correction coefficient k = 0.98 should be used. During measurements at −2 °C and with snow on the slab surface, a dielectric constant k = 1.02 correction coefficient should be adopted. During measurements at −2 °C and with de-icing salt on the slab surface, a dielectric constant correction factor k = 0.55 should be used. As a result of applied correction factors, measurement errors of thickness determinations have been reduced, especially if there is de-icing salt on the surface: up to −4 ÷ +9% in the case of the surfaces covered with de-icing salt (see Table 12). In other cases, after applying correction factors, the thickness measurement error was up to ±4% in reference conditions, up to ±10% in the case of the surface below zero temperature, up to −7 ÷ +11% in the case of the surface covered with water film, up to −6 ÷ +9% in the case of icy surfaces, and up to −17 ÷ +8% in the case of the surfaces covered with a thin layer of snow. It is noted that the coefficients have been calculated as the average of the coefficients for 5 slabs with different HMA wearing courses. To significantly reduce the error, a factor calculated for a specific HMA wearing course should be used.

Such coefficients have been proposed for specific slab geometry (m1–m5), specific measurement conditions (wI–wVI), and selected antenna (1.0 GHz air-coupled). It is proposed to continue the research in order to build a base of dielectric constant values of various asphalt mixtures determined at different temperature and humidity conditions based on the amplitude of the wave reflected from the asphalt layer and the corresponding correction factors, thus increasing the accuracy of the thickness determination of the layer by the GPR method and reducing the number of cores necessary to achieve the desired accuracy of thickness determination.

## 6. Conclusions

Based on the research and analysis carried out, the following was concluded:The value of dielectric constant that is necessary to determine the layer thickness based on the GPR method can be determined either based on the thickness of the cores (“B” method) or based on the amplitude of the wave reflected from the surface (“A” method). The “B” method is accurate, and a possibility of coring is usually limited; therefore, this method can be used for testing a homogeneous section in terms of the layer’s material. The “A” method allows recognizing the thickness over a large distance without drilling. Its accuracy depends on the surface conditions that result from atmospheric conditions that are present during the test, and they require a proper correction of dielectric constant.The value of the dielectric constant was calculated based on the amplitudes estimated in different atmospheric conditions: (changes in the temperature, moisture, presence of ice/snow, presence of de-icing salt). In the conducted test, the error of thickness measurement was:
(a)up to 11% when measuring in negative ambient temperature(b)up to 11% during measurements of the asphalt’s slab surface covered with water film(c)up to 8% during measurements of the ice-covered asphalt’s slab surface(d)up to 16% when measuring asphalt slab with a surface covered with a thin layer of fresh snow(e)even 29% during measurements of asphalt slab with a surface covered with a layer of de-icing salt
The purpose of this work was to improve the accuracy of determining the thickness of layers based on dielectric constants calculated based on the wave amplitudes reflected from the surface. It was proposed to achieve it using wavelet analysis and based on the wavelet scalogram to determine the effect of surface conditions during the measurements. Then, the correction factor for the dielectric constant was calculated based on the amplitude of the wave reflected from the surface, and the thickness was calculated based on the corrected dielectric constant value. After applying correction factors for dielectric constants, the error in determining the thickness was reduced (especially the error in measuring the thickness of the surface covered with de-icing salt). The average thickness determination errors were reduced:
(a)when measuring in negative ambient temperature, to on average 4%(b)when measuring the asphalt’s slab surface covered with water film, to on average 5%(c)when measuring the ice-covered asphalt’s slab surface to on average 4%(d)when measuring the asphalt slab with surface covered with a thin layer of fresh snow, to on average 6%(e)when measuring the asphalt slab with a surface covered with a layer of de-icing salt, to on average 4%
It should be noted that the correction coefficients have been calculated as the average of the coefficients for 5 slabs with different HMA wearing courses for specific atmospheric conditions. To significantly reduce the error, a dielectric constant correction factor calculated for a specific HMA wearing course should be applied.The usefulness of wavelet analysis for determining the presence of de-icing salt on the tested surface has been demonstrated—its presence is clearly visible on the wavelet scalogram. This is particularly useful, because the salt significantly affects the value of the dielectric constant and causes a thickness measurement error 21% (in average)—and it is not possible to recognize the presence of de-icing salt on the surface neither based on the visual assessment of the surface, nor based on basic radargram analysis. No significant changes were observed on the wavelet scalograms from the signal recorded in the different surface conditions.Until the time of satisfactory development of the correction coefficients for dielectric constants, it is recommended to use the hybrid method for determining the thickness of layers using the GPR method—calculating the dielectric constant based on the amplitude of the wave reflected from the surface and each time determining the correction coefficient due to the surface condition based on drilled cores. In addition, selective wavelet analysis is recommended for the possible detection of the presence of de-icing salt when measurements are taken in the autumn and winter.

## Figures and Tables

**Figure 1 materials-13-03214-f001:**
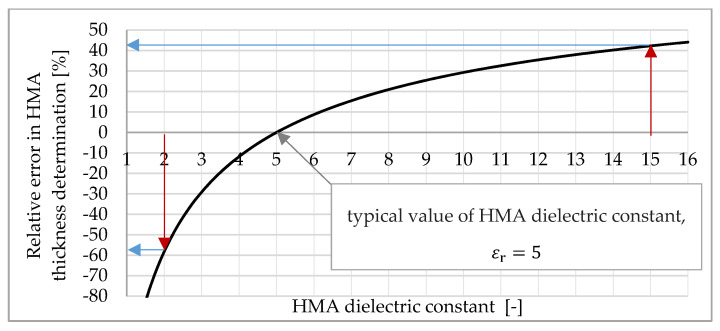
Impact of incorrect estimation of the dielectric constant of the asphalt mixture on the accuracy of thickness determination using the ground penetrating radar (GPR) method; blue arrows show the relative error in determining hot mix asphalt (HMA) thickness in situations when for HMA (22 cm thick and with an actual dielectric constant of 5), we incorrectly assume $\varepsilon_{r}=2$ or $\varepsilon_{r}=15$ (red arrows, the smallest/largest published HMA dielectric constant in the literature [18,20]).

**Figure 2 materials-13-03214-f002:**
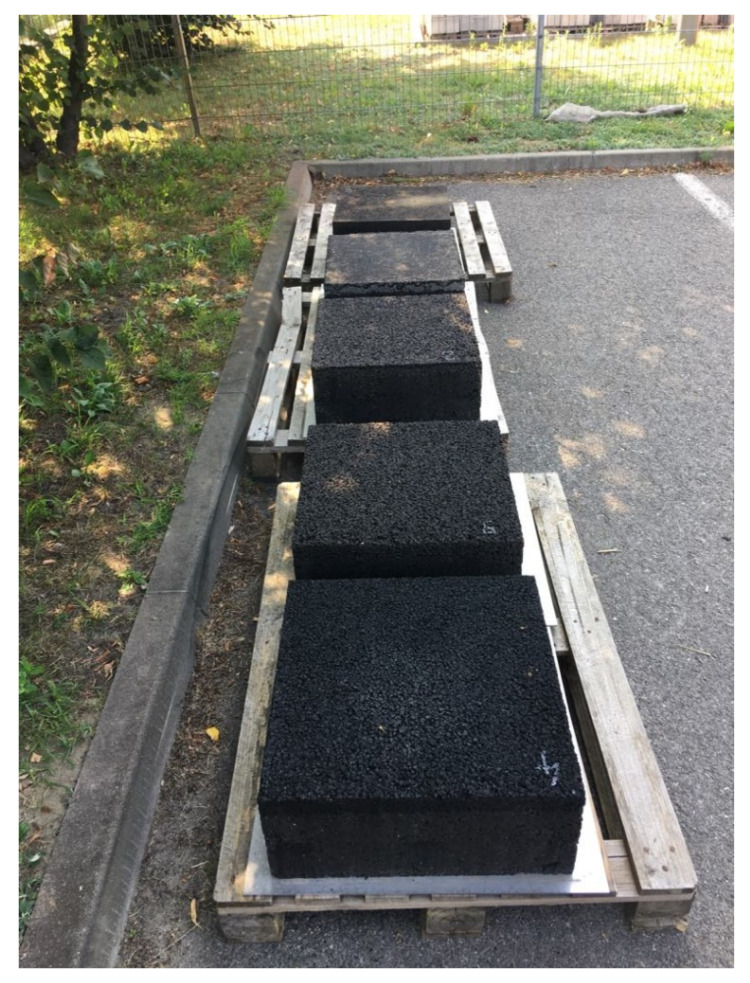
Tested model asphalt slabs.

**Figure 3 materials-13-03214-f003:**
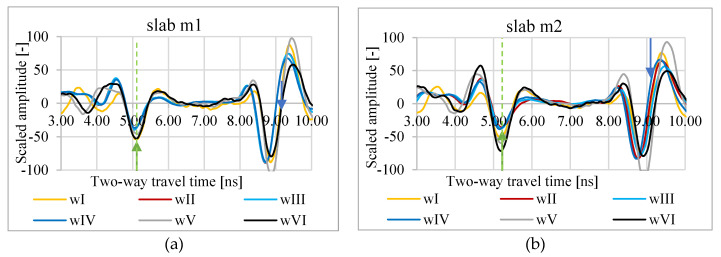
A-scans of m1 (**a**) and m2 (**b**) slabs in atmospheric conditions wI, wII, wIII, wIV, wV, and wVI; green arrow—assumed zero level, blue arrow—assumed bottom of the slab.

**Figure 4 materials-13-03214-f004:**
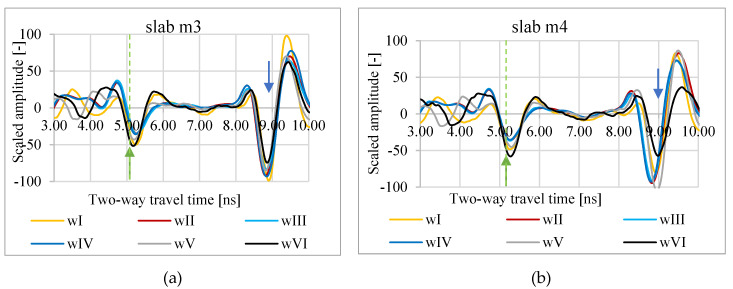
A-scans of m3 (**a**) and m4 (**b**) slabs in atmospheric conditions wI, wII, wIII, wIV, wV, and wVI; green arrow—assumed zero level, blue arrow—assumed bottom of the slab.

**Figure 5 materials-13-03214-f005:**
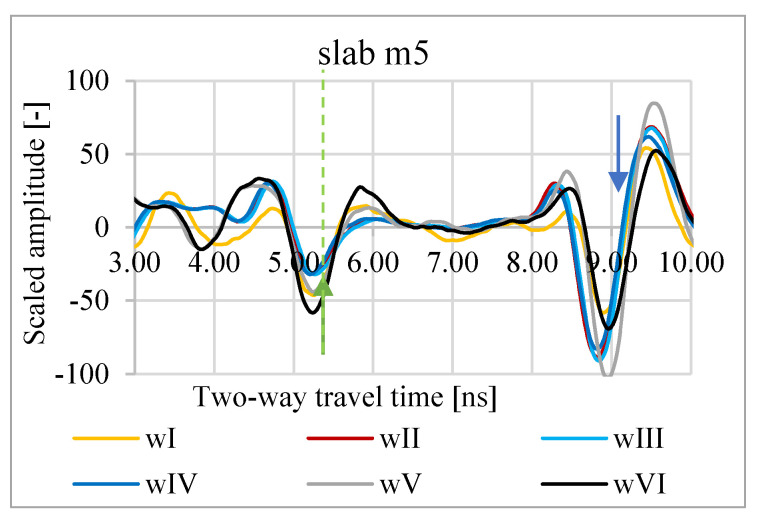
A-scans of m5 slab in atmospheric conditions wI, wII, wIII, wIV, wV, and wVI; green arrow—assumed zero level, blue arrow—assumed bottom of the slab.

**Figure 6 materials-13-03214-f006:**
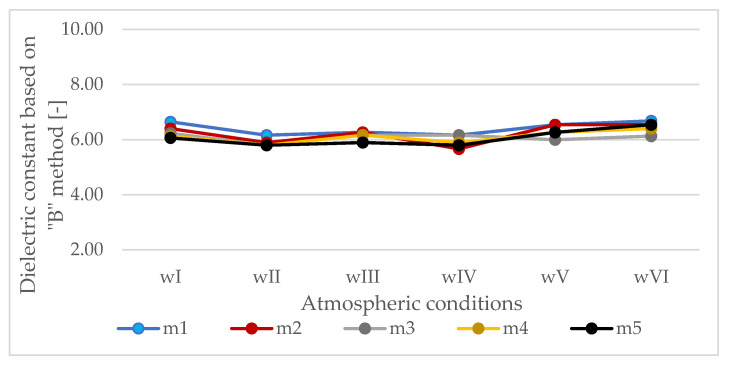
Dielectric constants calculated based on propagation time through the slabs (“B” method).

**Figure 7 materials-13-03214-f007:**
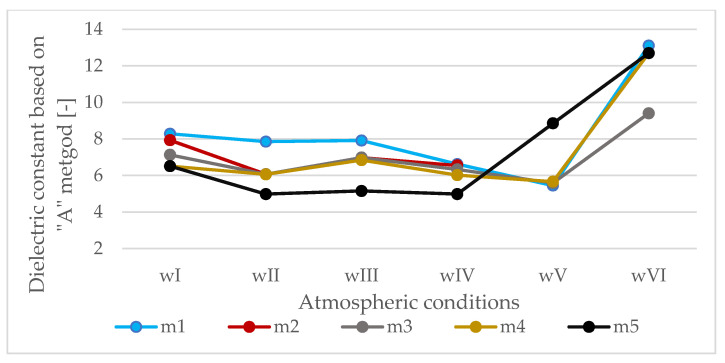
Dielectric constants calculated based on the amplitudes (“A” method).

**Figure 8 materials-13-03214-f008:**
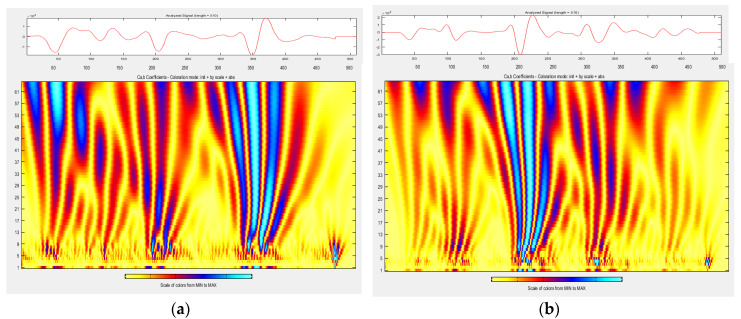
Wavelet analysis of A-scan from measurements in conditions (**a**) wI (temp. 28 °C, dry slab surface); (**b**) wII (temp. −5 °C, dry slab surface).

**Figure 9 materials-13-03214-f009:**
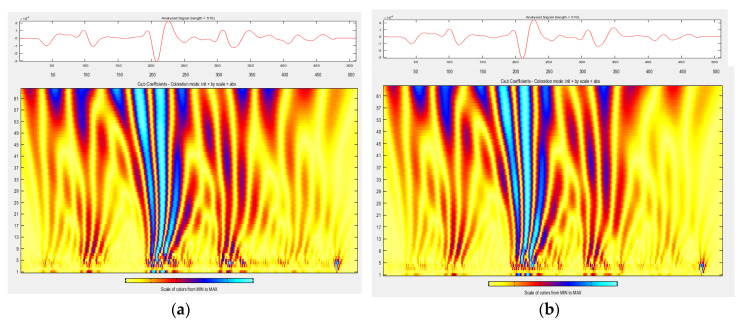
Wavelet analysis of A-scan from measurements in conditions (**a**) wIII (temp. −5 °C, water film on the slab surface); (**b**) wIV (temp. −5 °C, thin layer of ice on the slab surface).

**Figure 10 materials-13-03214-f010:**
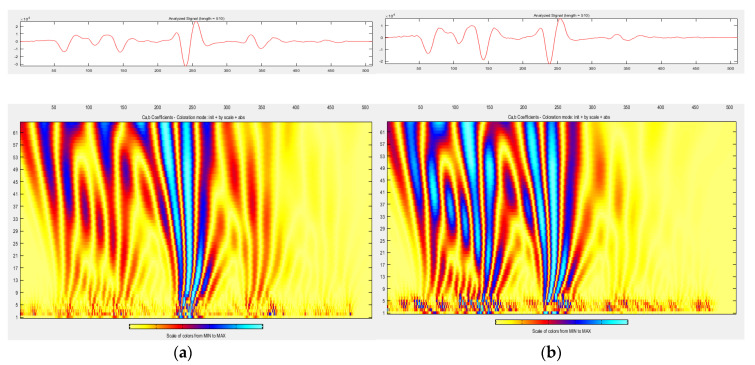
Wavelet analysis of A-scan from measurements in conditions (**a**) wV (temp. −2 °C, thin layer of fresh snow on the slab surface); (**b**) wVI (temp. −2 °C, de-icing salt on the slab surface).

**Table 1 materials-13-03214-t001:** Dielectric constants of asphalt mixtures published in the literature.

Dielectric Constant [-]	Note
2–4	value given as “typical” for asphalt [18]
2–4	dry asphalt [17]
6–12	wet asphalt [17]
2.5–3.5	value given as “typical” for asphalt [21]
3–6	value given as “typical” for asphalt [22]
4.0–4.9	value determined based on SMA (stone mastic asphalt) field tests, dielectric constant calculated based on the reflected wave amplitudes [23]
4–8	value given as “typical” for asphalt [20]
8–15	slag asphalt [20]
4–10	value given as “typical” for asphalt [19]

**Table 2 materials-13-03214-t002:** Constructions of model asphalt slabs.

Slab No.	Wearing Course (4 cm)	Bonding Layer (8 cm)	Base Layer (10 cm)
m1	MA 8	AC 16 W	AC 22 P
m2	SMA 8		
m3	AC 8 S		
m4	BBTM 8		
m5	PA 8 S		

**Table 3 materials-13-03214-t003:** Atmospheric conditions during GPR surveys.

Condition No.	Atmospheric Condition Description
wI	temperature 28 °C, dry slab surface
wII	temperature −5 °C, dry slab surface
wIII	temperature −5 °C, water film on the slab surface as a result of pouring water (about 5 L per slab)
wIV	temperature −5 °C, thin layer of ice on the slab surface
wV	temperature −2 °C, thin layer of fresh snow on the slab surface
wVI	temperature −2 °C, de-icing salt on the slab surface

**Table 4 materials-13-03214-t004:** Two-way travel time through the slabs.

*t* [ns]	m1	m2	m3	m4	m5
wI	3.78	3.71	3.66	3.63	3.61
wII	3.64	3.56	3.53	3.53	3.53
wIII	3.67	3.67	3.64	3.64	3.56
wIV	3.64	3.49	3.64	3.56	3.53
wV	3.75	3.75	3.59	3.67	3.67
wVI	3.79	3.75	3.63	3.71	3.75

**Table 5 materials-13-03214-t005:** Dielectric constants calculated based on propagation time through the slabs (‘B’ method).

$\varepsilon_{r_{B}}\;\left[-\right]$	m1	m2	m3	m4	m5	$\mathrm{Average}\;\varepsilon_{r_{B}}\;\left[-\right]$
wI	6.64	6.40	6.23	6.13	6.06	6.29
wII	6.16	5.89	5.79	5.79	5.79	5.88
wIII	6.26	6.26	6.16	6.16	5.89	6.15
wIV	6.16	5.66	6.16	5.89	5.79	5.93
wV	6.54	6.54	5.99	6.26	6.26	6.32
wVI	6.68	6.54	6.13	6.40	6.54	6.46

**Table 6 materials-13-03214-t006:** Ratio of the wave amplitude reflected from the surface of the slab to the wave amplitude reflected from the metal plate.

$A_{0}/A_{m}$ [−]	m1	m2	m3	m4	m5
wI	0.48	0.48	0.46	0.44	0.42
wII	0.47	0.45	0.42	0.42	0.38
wIII	0.48	0.45	0.43	0.45	0.39
wIV	0.44	0.44	0.43	0.42	0.38
wV	0.40	gross error	0.41	0.41	0.50
wVI	0.57	gross error	0.51	0.56	0.63

**Table 7 materials-13-03214-t007:** Dielectric constants calculated based on the amplitudes (“A” method’).

$\varepsilon_{r_{A}}\;\left[-\right]$	m1	m2	m3	m4	m5	$\mathrm{Average}\;\varepsilon_{r_{A}}\;\left[-\right]$
wI	8.28	7.93	7.13	6.51	6.51	7.27
wII	7.85	6.07	6.07	6.06	4.98	6.21
wIII	7.91	6.98	6.98	6.84	5.15	6.77
wIV	6.62	6.55	6.33	6.02	4.98	6.10
wV	5.45	gross error	5.59	5.66	8.85	6.39
wVI	13.10	gross error	9.40	12.69	12.69	11.97

**Table 8 materials-13-03214-t008:** Calculated slabs thicknesses based on the wave amplitude reflected from the surface.

d [cm]	m1	m2	m3	m4	m5
wI	19.70	19.76	20.56	21.34	21.22
wII	19.49	21.67	21.49	21.51	23.73
wIII	19.57	20.84	20.67	20.88	23.53
wIV	21.22	20.45	21.70	21.76	23.73
wV	24.09	gross error	22.78	23.14	18.50
wVI	15.71	gross error	17.76	15.62	15.79

**Table 9 materials-13-03214-t009:** Relative thickness determinations error caused by calculating the dielectric constant based on amplitudes.

Δd **[%]**	m1	m2	m3	m4	m5	Mean of the Absolute Value
wI	−10	−10	−7	−3	−4	7
wII	−11	−1	−2	−2	8	5
wIII	−11	−5	−6	−5	7	7
wIV	−4	−7	−1	−1	8	4
wV	10	gross error	4	5	−16	7
wVI	−29	gross error	−19	−29	−28	21

**Table 10 materials-13-03214-t010:** Correction coefficients for dielectric constants determined based on the wave amplitude reflected from the surface.

k [−]	m1	m2	m3	m4	m5	Average k [−]
wI	0.80	0.81	0.87	0.94	0.93	0.87
wII	0.78	0.97	0.95	0.96	1.16	0.97
wIII	0.79	0.90	0.88	0.90	1.14	0.92
wIV	0.93	0.86	0.97	0.98	1.16	0.98
wV	1.20	gross error	1.07	1.11	0.71	1.02
wVI	0.51	gross error	0.65	0.50	0.52	0.55

**Table 11 materials-13-03214-t011:** Calculated slabs thicknesses based on the corrected dielectric constant.

$d_{cor}\;\left[cm\right]$	m1	m2	m3	m4	m5
wI	21.1	21.2	22.0	22.9	22.7
wII	19.8	22.1	21.9	21.9	24.1
wIII	20.4	21.7	21.5	21.7	24.5
wIV	21.4	20.6	21.9	22.0	23.9
wV	23.8	gross error	22.5	22.9	18.3
wVI	21.3	gross error	24.1	21.2	21.4

**Table 12 materials-13-03214-t012:** Relative thickness determinations error after correction of dielectric constant.

Δd **[%]**	m1	m2	m3	m4	m5	Mean of the Absolute Value
wI	−4	−4	0	4	3	3
wII	−10	0	−1	−1	10	4
wIII	−7	−1	−2	−1	11	5
wIV	−3	−6	0	0	9	4
wV	8	gross error	2	4	−17	6
wVI	−3	gross error	9	−4	−3	4

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
