# Peer review of "The Use of Wavelet Analysis to Improve the Accuracy of Pavement Layer Thickness Estimation Based on Amplitudes of Electromagnetic Waves"

_materials, 2020, doi:10.3390/ma13143214_

Round 1
Reviewer 1 Report
The manuscript concerns the use of wavelet analysis to improve the accuracy of pavement layer thickness estimation based on amplitudes of electromagnetic waves. The research concludes that the value of the dielectric constant, based on the amplitude of the wave reflected from the surface, is affected by some parameters related with the atmospheric conditions on the surface. The work is very interesting, sufficiently innovative, well organized and described. An additional check of the English language and style is advisable.
Some minor remarks:
23. Introduction contains three references only. It is not clear if the description of existing techniques refers to these articles or also to others. In any case, this section should be enlarged.
Table 1. Put the reference next to each note and repeat it, if it is the same (e.g. dry and wet pavement)
86. The methods ... is= consider to
113. 42=+42
Figure 1 caption: blue arrows show... Maybe not all the arrows.
177. 25 MHz, 50 MHz, 250 MHz and 500 MHz = 25, 50, 250 and 500 MHz
351. Discussion should be followed by conclusions. If they coincide, conclusions should be shortened.
Author Response
Please see the attachment. In response to Your comments, the following has been changed / supplemented:
24-39. Several items with references were added to the Introduction.
Table 1. References have been put next to each note.
86.The methods consider to...
115.+42%
Figure 1. Impact of incorrect estimation of the dielectric constant of the asphalt mixture on the accuracy of thickness determination using the GPR method; blue arrows show the relative error in determining HMA thickness in situation when for HMA (22 cm thick and with an actual dielectric constant of 5) we incorrectly assume Er=2 or Er=15 (red arrows, the smallest/largest published HMA dielectric constant in the literature [18,20]).
181. In measurements taken with 25, 50, 250 and 500 MHz ground-coupled antennas to distinguish ...
432-485. Conclusions

Reviewer 2 Report
Referee’s comments on " The use of wavelet analysis to improve the accuracy of pavement layer thickness estimation based on amplitudes of electromagnetic waves "
The aim of this paper is an estimation of pavement thickness with GPR based on advanced signal analysis using wavelet transform. Based on tests on model asphalt slabs, it was presented how the value of dielectric constant changes depending on the atmospheric conditions of the measured surface (dry, covered with water film, covered with ice, covered with snow, covered with de-icing salt). The tests and their results are interesting. However, the paper is not well-written.
Some comments are given as follow.
1)There is no section of conclusions in the paper. Please add it.
2)Why can the accuracy of pavement layer thickness estimation be improved based on amplitudes of electromagnetic waves by using of wavelet analysis? Give a summarize of the reasons in the text.
3)Please provide the photo of slab.
Author Response
Please see the attachment. In response to Your comments, the following has been changed / supplemented:
1) 434-485. Conclusions are provided.
2) 452-467. The purpose of this work was to improve the accuracy of determining the thickness of layers based on dielectric constants calculated based on the wave amplitudes reflected from the surface. It was proposed to achieve it using wavelet analysis and based on the wavelet scalogram to determine the effect of surface conditions during the measurements. Then, the correction factor for the dielectric constant was calculated based on the amplitude of the wave reflected from the surface and the thickness was calculated based on the corrected dielectric constant value. After applying correction factors for dielectric constants, the error in determining thickness was reduced (especially the error in measuring the thickness of the surface covered with de-icing salt). The average thickness determination errors were reduced.
3) 204. Photo of tested model asphalt slabs is provided.

Reviewer 3 Report
I have several comments:
- There is no reference added to equation (1-3). Equations are suggested by manufacturer, authors or some one else?
- Line 106 "... 3,3 ns", not clear, what is that "ns"?
Author Response
Please see the attachment. In response to Your comments, the following has been changed / supplemented:
- Equation (1) - [5], Equation (2) - [5,6], Equation (3) - [25].
[5] Lalagüe A., Use of Ground Penetrating Radar for Transportation Infrastructure Maintenance, Thesis for the Degree of Philosophiae Doctor at Norwegian University of Science and Technology, Trondheim, 2015, pp. 1-190.
[6] Li, W.; Wen, J.; Xiao, Z.; Xu, S. Application of Ground-Penetrating Radar for Detecting Internal Anomalies in Tree Trunks with Irregular Contours, Sensors, 2018, 18, 649. doi: 10.3390/s18020649.
[25] Iyengar B., A textbook of Engineering Mathematics, Laxmi Publications, 2004, ISBN-10: 8170083656,
1425. - Line 106: The two-way travel time is then 3,3 ns (nanoseconds).

Round 2
Reviewer 2 Report
The authors have done a good job in addressing my comments. The manuscript is acceptable to me.